behaviour/genetics

rice farming, agriculture, genetic adaptation, polygenic scores, China

**Author for correspondence:**
Chen Zhu
e-mail: zhuchen@cau.edu.cn

# Relationship between rice farming and polygenic scores potentially linked to agriculture in China

Chen Zhu[1,2,3,†], Thomas Talhelm[4,†], Yingxiang Li[5], Gang Chen[5,6], Jiong Zhu[7,8] and Jun Wang[9]

[1]College of Economics and Management, [2]Academy of Global Food Economics and Policy (AGFEP), and [3]Beijing Food Safety Policy and Strategy Research Base, China Agricultural University, Beijing 100081, People's Republic of China
[4]Booth School of Business, University of Chicago, Chicago, IL 60637, USA
[5]WeGene, Shenzhen Zaozhidao Technology Co. Ltd, Shenzhen, People's Republic of China
[6]Hunan Provincial Key Lab on Bioinformatics, School of Computer Science and Engineering, Central South University, Changsha, People's Republic of China
[7]Institute of Economics, School of Economics, and [8]Wang Yanan Institute for Studies in Economics (WISE), Xiamen University, Xiamen, People's Republic of China
[9]School of Public Administration and Policy, Renmin University of China, Beijing, People's Republic of China

(iD) CZ, 0000-0003-3582-9022

Following domestication in the lower Yangtze River valley 9400 years ago, rice farming spread throughout China and changed lifestyle patterns among Neolithic populations. Here, we report evidence that the advent of rice domestication and cultivation may have shaped humans not only culturally but also genetically. Leveraging recent findings from molecular genetics, we construct a number of polygenic scores (PGSs) of behavioural traits and examine their associations with rice cultivation based on a sample of 4101 individuals recently collected from mainland China. A total of nine polygenic traits and genotypes are investigated in this study, including PGSs of height, body mass index, depression, time discounting, reproduction, educational attainment, risk preference, *ADH1B* rs1229984 and *ALDH2* rs671. Two-stage least-squares estimates of the county-level percentage of cultivated land devoted to paddy rice on the PGS of age at first birth ($b = -0.029$, $p = 0.021$) and *ALDH2* rs671 ($b = 0.182$, $p < 0.001$) are both statistically significant and robust to a wide range of potential confounds and alternative explanations. These findings imply that rice farming may influence human evolution in relatively recent human history.

†These authors contributed equally to this work.

# 1. Introduction

Agriculture was one of the most critical transitions in human history. It fundamentally changed the way people live [1]. Yet, not all farming is the same. Rice farming is particularly important because it was the foundation of some of the world's largest civilizations. Over half the world's population (53%) lives in societies with significant legacies of rice farming [2]. Rice is also essential because it was so radically different from other common staple crops [3]. Although rice is not the only grain crop that humans have relied on, it is different from other major staple cereals such as wheat, barley and millet in important ways. Pre-modern paddy rice required about twice the labour hours as wheat and millet [3–6]. Furthermore, because it grows best in standing water, rice farmers often had to manage shared irrigation systems. Those shared systems often forced farmers to coordinate their water use and sometimes even flood and drain their fields at the same time [3,6]. Thus, labour and irrigation made rice farmers more dependent on each other.

There is some observational evidence that rice cultures differ from nearby cultures that farm other crops. For example, Davidson [7] found that people in rice-farming societies had a particularly strong work ethic. Talhelm *et al.* [8] reported that people from rice-farming areas are more interdependent than people from wheat-farming areas. People in rice areas of China were less likely to be alone and even less likely to move a chair out of their way in Starbucks—a sort of fitting into the environment that is more common in interdependent societies [9]. Around the globe, societies with a history of rice farming have less 'relational mobility' [10]. Low-mobility societies have more secure, long-term relationships, but less flexibility and fewer opportunities to meet new people. However, this literature has only analysed the relationship between rice farming and phenotypic traits. We know very little about the mechanisms of how rice farming leads to behaviours. This is particularly a puzzle when we consider evidence of rice–wheat differences among middle-class customers in Starbucks—people who have presumably never farmed rice or wheat in their lives. One possibility is that genes play a role in carrying on behavioural differences between people from rice cultures and non-rice cultures. In this study, we leverage a unique dataset recently collected from counties across China and findings from development in molecular genetics to test whether pre-modern rice farming is associated with modern polygenic traits.

## 1.1. Three reasons rice-farming genetic selection may be plausible

Although the idea that behavioural differences resulting from rice cultivation history may be partly genetic seems controversial, three lines of findings support the possibility. First is the idea that human evolution is not limited to the distant past. Recent research in evolutionary biology has found evidence that natural selection is still operating in contemporary humans [11–13]. To name a few examples, researchers have found evidence for natural selection on height, waist-to-hip ratio, skin colour, spleen size and infant head circumference within the last few thousand years [14–16]. What is more, researchers have demonstrated that individual genotyping data can be used to directly measure the action of selection [13,17].

Second, there is accumulating and converging biological and genetic evidence to show that the transition from hunter–gatherer to agricultural societies exerted selective pressures on human evolution [16,18–21]. A well-known example is a genetic adaptation to digest lactose from milk in adulthood. Studies have found this adaptation came after some human groups domesticated dairy animals 8000 years ago [19]. Researchers have found compelling evidence that cattle domestication and dairying actively selected for lactose-tolerant genes among some Neolithic humans but not others [18–21]. There are yet more examples of genetic changes linked to the agricultural revolution. Mathieson *et al.* [21] analysed the ancient DNA of Europeans (who lived between 6500 BC to 300 BC) and found evidence of genetic changes in height, digestion and the immune system that were probably adaptations to settled agricultural life. Other researchers have documented archaeological evidence that early humans' bone density decreased after humans started agriculture [22,23]. These findings raise the possibility that rice farming—a unique agricultural practice that has been around for thousands of years—might have nudged genetic variations in certain traits.

The third line of thought is the accumulating evidence linking behavioural and personality traits to genes [24,25]. Much of this evidence has come from genome-wide association studies (GWAS). Scientists have identified genetic factors associated with reproductive preferences [26], risk preferences [27], time discounting [28] and educational attainment [29]. Given that genetics influence a wide range of

behaviours, it is plausible to hypothesize that they influence—even if in a small way—differences between rice-farming cultures and wheat-farming cultures.

## 1.2. China as a natural test case

The current study builds on existing GWAS evidence and leverages a unique Chinese dataset to test for selection by rice domestication and cultivation. China presents a unique test case for the theory for three reasons. First, China has a long history of rice cultivation, giving it enough time for genetic selection to play out [30]. Second, China spans a large geographical area, with millions of people in traditionally rice-farming areas and millions of people in non-rice-farming areas. That gives enough statistical power to test the theory in a robust way. Third, despite China's large population and landmass, it is relatively unified in terms of politics, religion and language (especially compared with other areas with similar population sizes, such as sub-Saharan Africa or the Indian subcontinent). That makes it is easier—but surely still with some difficulty—to limit confounds in ethnicity, national government and language.

## 1.3. Trait candidates

Out of all the possible behaviours and genes that rice might have affected, where to start? We start with a few plausible candidates for phenotypes that might be connected to rice. The advent of agriculture reshaped diet, patterns of labour, population density and settled living [20,23]. Because this is an initial exploration of the theory, we test a broad range of physiological and behavioural traits. Nevertheless, we constrain the set to phenotypes that fit three criteria: (i) factors that might plausibly be linked to diet and subsistence style, (ii) factors that have received multiple empirical verifications linking genes to phenotype, and (iii) factors that give a broad coverage of physiological differences and behavioural traits [26,28–33]. This resulted in genetic variants for bodily dimensions (height and BMI), mental health (risk of depression), alcohol metabolism capacity (alcohol dehydrogenase 1B or *ADH1B* rs1229984, and aldehyde dehydrogenase 2 or *ALDH2* rs671), economic preferences (time, risk and reproductive preferences) and socioeconomic outcome (educational attainment).

We began by constructing several polygenic scores (PGSs) to measure these genome-wide complex traits. The sample consisted of 4101 adult participants from WeGene who consented to participate in research (see Methods). For each individual, we summed the weights of all related alleles. These PGSs are aggregated effects of hundreds or thousands of trait-associated DNA variants identified in GWAS studies, and can be used to predict propensities towards certain traits and outcomes [13,17]. Table 1 presents summary statistics of key variables in the sample.

## 1.4. The measure of rice cultivation

The main explanatory variable is defined as the percentage of cultivated land per county devoted to paddy rice (in a total of 328 counties from 30 provinces; figure 1). We use the earliest county-level rice data we could find, which for most provinces is around the year 2000. These recent statistics correlate highly with rice data from a more limited dataset from 1914 to 1918, $r = 0.95$, $p < 0.001$ [37]. Thus, the rice statistics seem to adequately represent historical rice farming. For simplicity, we describe non-rice-farming regions as 'wheat-farming' regions. This is a simplification because non-rice regions also traditionally grew similar dryland crops like millet and barley [3,35]. But, on the whole, rice is negatively correlated with wheat in China $r_{prov} = -0.69$, $p < 0.001$.

# 2. Results

## 2.1. Basic rice–wheat differences

Figure 2 plots the average PGSs (or genotypes) against the county-level percentage of cultivated land devoted to paddy rice, with linearly fitted values in solid blue lines. At first glance, several behavioural and physiological polygenic traits are correlated to the intensity of paddy rice farming, such as the PGS of height (figure 2, (1), decreasing), *ADH1B* rs1229984 (figure 2, (3), increasing), *ALDH2* rs671 (figure 2, (4), increasing), the PGS of age at first birth (figure 2, (7), decreasing), etc.

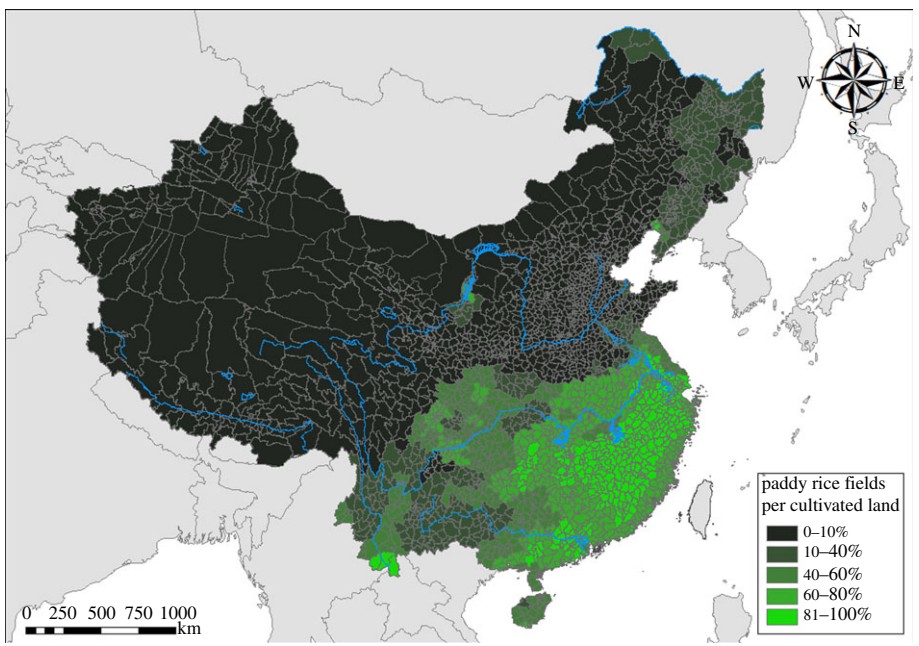

**Figure 1.** County percentage of cultivated land devoted to paddy rice. County-level paddy rice data come from the year 2002 Statistical Yearbook. However, analyses comparing data from 1914–1918 with modern statistics find that modern statistics correlate highly with historical farming data.

**Table 1.** Summary statistics of the analytical sample ($N = 4101$). PGSs of educational attainments, depression, time discounting, age at first birth, height, BMI and risk tolerance are normalized from 0 to 1.

| variable | mean (s.d.) | range |
|---|---|---|
| socio-demographic characteristics | | |
| age | 27.4 (7.4) | 18–67 |
| years of schooling | 16.2 (2.3) | 6–22 |
| urbanization of youth environment | countryside: 9.1%; town: 41.8%; city: 49.1% | |
| birthplace characteristics | | |
| county per cent paddy rice field | 0.462 (0.354) | 0.000–0.952 |
| county 2012 GDP *per capita* (10 000 CNY) | 5.659 (2.771) | 0.646–16.301 |
| historical pathogen prevalence (province) | 14.4 (7.2) | 0.4–25.7 |
| county yearly average temperature (℃) | 15.3 (4.6) | −1.0–25.8 |
| polygenic scores (PGSs) and genotypes | | |
| educational attainment [33] | 0.53 (0.15) | 0–1 |
| depression [34] | 0.51 (0.13) | 0–1 |
| time discounting [32] | 0.84 (0.18) | 0–1 |
| age at first birth [30] | 0.53 (0.14) | 0–1 |
| height [35] | 0.52 (0.15) | 0–1 |
| BMI [36] | 0.51 (0.16) | 0–1 |
| risk tolerance [31] | 0.54 (0.13) | 0–1 |
| *ALDH2* rs671 | 0.38 (0.56) | 0–2 |
| *ADH1B* rs1229984 | 1.25 (0.73) | 0–2 |

## 2.2. Alternative theories and individual control variables

Nevertheless, differences that appear to be due to rice and wheat might actually be driven by a wide range of other regional differences. For example, rice grows in southern latitudes, and researchers

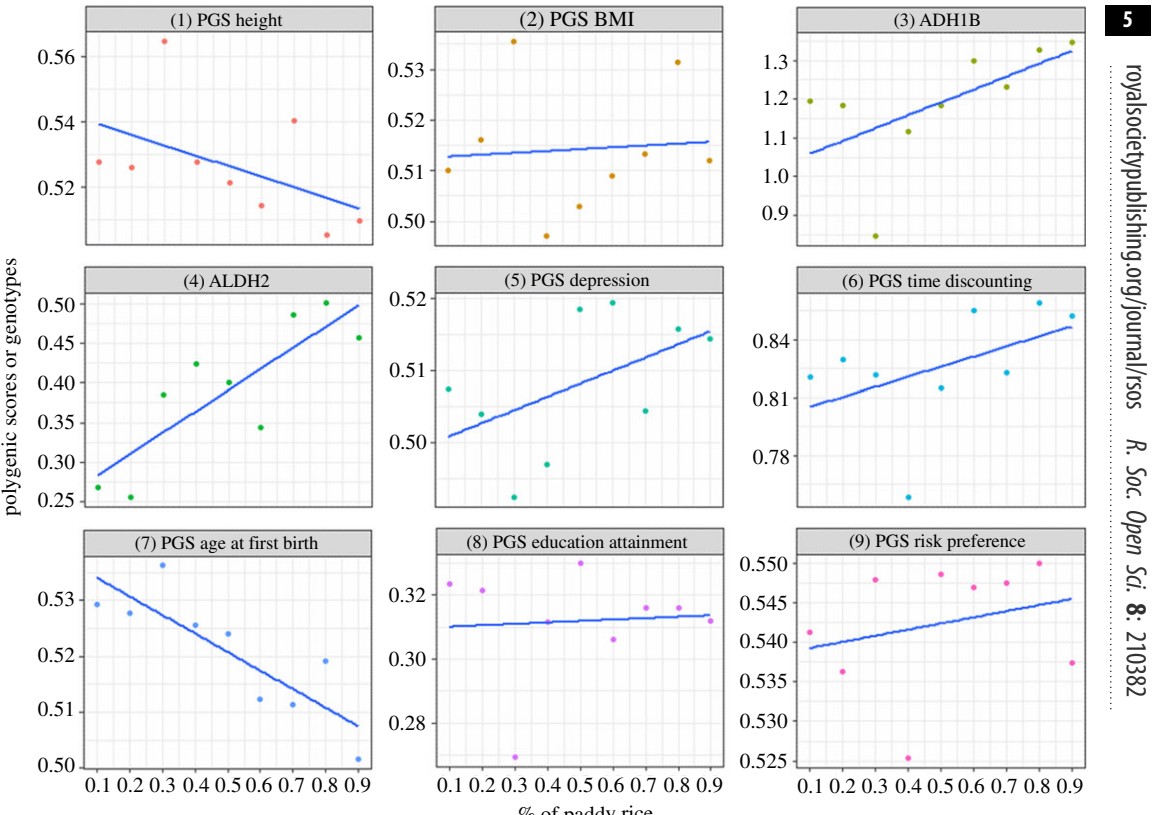

**Figure 2.** Average polygenic scores by county rice paddy percentage.

have reported that genetic differences follow a latitudinal gradient from north to south in China. Rice is also correlated with temperature, and, around the world, the temperature is correlated with both height and BMI, albeit weakly (known as Bergmann's rule).

To counteract this problem, we run multivariate regressions testing the effect of rice on polygenic traits while accounting for an extensive range of additional factors. First, we take into account participants' age and the size of the city they grew up in (rural/town/city). Then, based on their birthplace, we control for regional characteristics that might confound the relationship between rice and genes: latitude, longitude, temperature, GDP, distance to coast, contact with herding cultures, regional education, history of rebellion and length of rivers (table 2 lays out all regional variables, sources and theories). Since China includes different ethnic populations, such as Mongolians and Tibetans, we also take into account people's ethnic make-up. Based on this accounting of population stratification, we adjust the model by a total of 42 ancestral compositions for each individual. The 42 ancestries are estimated directly from each individual's genotyping data using the ADMIXTURE program, including (from most to least frequent): Northern Han, Southern Han, Mongolian, Japanese, Naxi/Yi, Dai, Gaoshan, Kinh, Korean, She, Tibetan, Tungus, Ashkenazi, Balkan, Bantusa, Bengali, Cambodian, Egyptian, English, Eskimo, Finnish/Russian, French, Hungarian, Iranian, Kyrgyz, Lahu, Mala, Mayan, Mbuti, Miao/Yao, Papuan, Pima, Sardinian, Saudi, Sindhi, Somali, Spanish, Thai, Uygur, Uzbek, Yakut and Yoruba.

Table 3A reports parameter estimates from separate multivariate regressions of each polygenic trait on the percentage of paddy rice. In the benchmark model (Model 1), we include only paddy rice, age, the city size of the place where participants grew up (rural, town or city), 42 ancestral composition variables and latitude/longitude. This model finds that paddy rice is significantly associated with four out of nine traits (PGSs of time discounting, age at first birth, educational attainment and genotype of *ALDH2* rs671).

We then sequentially add regional control variables: average temperature (Model 2), distance to coast (Model 3), GDP (Model 4), contact with herding cultures (Model 5) and history of rebellion (Model 6). In almost all models, rice significantly predicts *ALDH2* rs671 and people's age of reproduction (i.e. age at first birth PGS). These results fit the idea that selection from rice farming has been operating on genetic variants associated with these two traits. By contrast, the PGSs or genotypes of the other traits were not

**Table 2.** Description of regional variables.

| variable | measure | source | rationale |
|---|---|---|---|
| % paddy rice field | paddy fields area/total cultivated area | China Statistical Yearbook, 2002 | paddy rice required more work and coordination to build and operate irrigation systems |
| GDP | GDP *per capita* in 2012 | China Statistical Yearbook, 2013 | this measures regional economic development |
| % cultivated land | hectares of cultivated land/ total province land | China Statistical Yearbook, 1996 | this measures the density of farming in general |
| environmental rice suitability | environmental suitability for high-input rainfed rice | UN Global Agro-Ecological Zones Database | environmental rice suitability (regardless of whether people actually farm rice there) is an instrumental variable to test reverse causality—the possibility that areas that were collectivistic to begin with chose to farm rice [8] |
| contact with herding cultures | people from traditionally herding ethnicities/total population | China Population Statistical Yearbook, 2002 | research has found that herding cultures tend to be more individualistic than nearby farming cultures [34] |
| distance to coast | The distance of provincial capital to nearest coast (100 km). Coast province = 0. | Marine Regions Database | Distance from the coast can be a proxy for modern and historical development. Coastal provinces also had more access to sea transport and potentially more diverse ideas and cultures. |
| average temperature | average high, low temperatures in January and July | Zuzu Che Weather Records | some researchers have argued that hotter areas are more collectivistic, and the temperature is also correlated with disease prevalence [36] |
| latitude | average of northernmost and southernmost province latitude | Google Maps | In China, rice is highly correlated with latitude. Latitude is a proxy for other environmental factors such as temperature and disease. Testing latitude checks the robustness of rice against latitude. |
| longitude | average of easternmost and westernmost province longitude | Google Maps | testing longitude checks the robustness of rice against longitude |
| historical pathogen epidemics | rates of epidemic diseases in the Ming and Qing Dynasty (1368–1911) | [38] | pathogen prevalence theory argues that environments with higher rates of communicable disease tend to be more collectivistic [39] |
| history of rebellion | an index of the frequency of mass rebellions during the Qing Dynasty | [40] | rebellion may have affected genetic selection or it may reflect regional cultural differences |

**Table 3.** Estimates from separate regressions of each polygenic trait (or genotype) on the proportion of paddy rice. This table shows parameter estimates (with *p*-values) of the effect of rice and control variables from separate regressions of each polygenic trait (or genotype). All models include the covariates of age, the city size of the place where participants grew up (rural, town or city) and ethnic ancestry.

| outcome variables: polygenetic scores (or genotypes) | | (1) height | (2) BMI | (3) alcohol intolerance: ADH1B | (4) alcohol intolerance: ALDH2 | (5) depression | (6) time discounting | (7) age at first birth | (8) educational attainment | (9) risk preference |
|---|---|---|---|---|---|---|---|---|---|---|
| (A) multivariate regressions | | | | | | | | | | |
| Model 1: benchmark (controlling for ethnic ancestry and latitude/longitude) | β | −0.009 | 0.001 | 0.052 | 0.165*** | −0.006 | 0.023* | −0.017* | −0.012* | 0.000 |
| | *p*-value | 0.404 | 0.901 | 0.299 | <0.001 | 0.547 | 0.075 | 0.078 | 0.056 | 0.979 |
| Model 2: further controlling for temperature | β | −0.001 | 0.001 | −0.012 | 0.198*** | −0.005 | 0.021 | −0.015 | −0.015** | −0.007 |
| | *p*-value | 0.934 | 0.936 | 0.851 | <0.001 | 0.682 | 0.191 | 0.225 | 0.047 | 0.547 |
| Model 3: further controlling for distance from coast | β | −0.002 | −0.002 | −0.005 | 0.186*** | −0.004 | 0.024 | −0.020* | −0.018** | −0.008 |
| | *p*-value | 0.876 | 0.910 | 0.932 | <0.001 | 0.750 | 0.137 | 0.094 | 0.020 | 0.493 |
| Model 4: further controlling for GDP | β | 0.004 | −0.002 | −0.021 | 0.240*** | −0.006 | 0.011 | −0.032** | −0.021** | −0.003 |
| | *p*-value | 0.802 | 0.902 | 0.763 | <0.001 | 0.638 | 0.522 | 0.018 | 0.014 | 0.793 |
| Model 5: further controlling for contact with herding cultures | β | 0.006 | −0.003 | −0.220** | 0.263*** | −0.000 | 0.030 | −0.041** | −0.019 | 0.028 |
| | *p*-value | 0.788 | 0.875 | 0.029 | 0.001 | 0.994 | 0.244 | 0.037 | 0.124 | 0.125 |
| Model 6: further controlling for rebellion history | β | 0.003 | 0.009 | −0.136 | 0.251*** | 0.005 | 0.038 | −0.046** | −0.013 | 0.028 |
| | *p*-value | 0.894 | 0.704 | 0.198 | 0.002 | 0.789 | 0.158 | 0.025 | 0.331 | 0.157 |

(*Continued.*)

**Table 3.** (Continued.)

| | | (1) | (2) | (3) | (4) | (5) | (6) | (7) | (8) | (9) |
|---|---|---|---|---|---|---|---|---|---|---|
| outcome variables: polygenetic scores (or genotypes) | | height | BMI | alcohol intolerance: ADH1B | alcohol intolerance: ALDH2 | depression | time discounting | age at first birth | educational attainment | risk preference |
| (B) two-stage least squares (2SLS) | | | | | | | | | | |
| Model 7: control for endogeneity of rice farming | $\beta$ | −0.003 | −0.002 | 0.020 | 0.182*** | −0.003 | 0.010 | −0.029** | −0.015 | −0.002 |
| | $p$-value | 0.741 | 0.373 | 0.108 | <0.001 | 0.695 | 0.750 | 0.021 | 0.161 | 0.169 |
| | first stage F-statistic | 84.740 | 84.740 | 84.740 | 84.740 | 84.740 | 84.740 | 84.740 | 84.740 | 84.740 |
| | $p$-value | 0.000 | 0.000 | 0.000 | 0.000 | 0.000 | 0.000 | 0.000 | 0.000 | 0.000 |
| | Sargan statistic | 2.183 | 2.445 | 2.372 | 2.103 | 3.087 | 2.535 | 2.185 | 1.805 | 2.218 |
| | $p$-value | 0.140 | 0.118 | 0.124 | 0.147 | 0.079 | 0.111 | 0.139 | 0.179 | 0.136 |
| | $N$ | 4101 | 4101 | 4101 | 4101 | 4101 | 4101 | 4101 | 4101 | 4101 |

*** $p < 0.01$; ** $p < 0.05$; * $p < 0.10$.

robustly associated with rice farming. After controlling for other regional differences, rice was not significantly linked to educational attainment, time discounting or *ADH1B* rs1229984.[1]

## 2.3. Testing reverse causality

A potential threat to this analysis is reverse causality. In other words, perhaps certain people had genetic traits that were more suited to rice farming, so they chose to grow rice. For example, if the sensitivity to social norms helps solve the free-rider problem in collective irrigation systems [3], perhaps areas where people were already more sensitive to social norms were then more likely to start farming rice. In this way, genes would (in a sense) cause rice farming, rather than rice farming selecting for genes.

To test for reverse causality, we exploit exogenous variations that determine regional differences in rice farming. We select a natural instrumental variable (IV) that measures environmental suitability for wetland rice modelled by the United Nations Food Agriculture Organization's Global Agro-ecological Zones database [8]. Then we use two-stage least-squares (2SLS) models to tease apart the causal impact of rice farming on polygenic traits by incorporating the environmental suitability of rice growing variable and its quadratic form as instruments.

Table 3B presents 2SLS estimates for the causal influence of rice farming on polygenic traits (using the full set of control variables from Model 7). We first test the validity of these instruments using the first stage *F*-test and the overidentification test. In the first stage, the *F*-statistic is 84.74, far exceeding the traditional cut-off of 10 for weak instruments. The *p*-values of the Sargan statistics are all higher than 0.1, indicating that the environmental suitability variable and its quadratic form are strong and valid instruments for the percentage of paddy rice field.

Compared with the earlier rice farming results (table 3A), the results incorporating environmental rice suitability remain robust for the *ALDH2* rs671 genotype (aldehyde dehydrogenase deficiency) and age at first birth. Hence, our 2SLS estimation results imply that the selective pressures from paddy rice farming seem to favour individuals with a lower alcohol tolerance and people with polygenetic scores for having children at an earlier age. Furthermore, reverse causality is not likely to be driving the results.[2]

## 2.4. Additional robustness checks

We then extend our analysis by performing a number of checks to test whether the findings are robust. First, we test historical disease rates in order to test the pathogen prevalence theory. The pathogen prevalence theory argues that, in areas with more communicable diseases, humans developed behaviours that helped protect them against disease [39]. For example, researchers have found that areas with higher historical rates of disease have higher fertility rates, lower birth weights and higher collectivism [39]. Because diseases are so clearly linked to life and death, pathogens seem like a plausible place to look for factors that would influence genetic selection.

To measure historical pathogen prevalence, we use rates of epidemic diseases in the Ming and Qing Dynasty at the province level (AD 1368–1911) [38]. These data lack three outlying provinces (Qinghai, Xinjiang and Inner Mongolia), leaving a total of 3956 participants. Pathogen rates did not predict genetic differences (table 4A).

Second, we test whether farming in general (as opposed to rice farming in particular) can explain genetic differences. To measure farming, we use the percentage of cultivated land per province. The multivariate regression results show that farming does not predict genetic differences, except for genetic variants associated with educational attainment (table 4B). Why might farming density select for educational attainment? This link may imply selection on specific brain functions or non-cognitive traits, which are correlated with genetic components of educational attainment [13,29]. Another theory

---

[1]The fact that we consider nine different outcomes raises concerns about multiple hypothesis testing (MHT) and false positives. Failure to account for MHT can lead to a substantial risk of false discoveries in the econometric analysis [41]. In order to address the issue, we implement a common *p*-value correction method of the D/AP procedure and evaluate adjusted *p*-values for robustness [42]. For correlated $N$ outcome variables to be tested, the adjusted *p*-value is calculated as $p_{1,\text{adjusted}} = 1 - (1 - p_1)^{M1}$, where $M1 = N^{1-r1}$, $r1 = (N-1)^{-1}\sum_{j \neq n}^{N} r_{jn}$, and $r_{jn}$ is the correlation coefficient between the $j$th and $n$th outcome variables, and so forth. Additional results after accounting for MHT are generally consistent with main findings presented in Table 3, and are available upon request from the corresponding author.

[2]To relax the assumption of strict exogeneity of instrumental variables, we also employ the plausibly exogenous approach built upon the inference strategy of plausibly exogenous instruments [43,44]. Plausibly exogenous estimation results suggest that violations to the exclusion restriction assumption may not be a critical issue in the current study. Results are available upon request from the corresponding author.

**Table 4.** Testing alternative theories. This table shows parameter estimates (with *p*-values) for two alternative theories: historical pathogen prevalence (A) and the proportion of land that is cultivated, which tests farming in general, as opposed to rice farming in particular (B). All models include age, urbanization, 42 ethnic ancestry variables, latitude, longitude, temperature, distance to coast, GDP, history of herding and history of rebellion.

| outcome variables: polygenetic scores (or genotypes) | (1) height | (2) BMI | (3) ADH | (4) ALDH | (5) depression | (6) time discounting | (7) age at first birth | (8) educational attainments | (9) risk preference |
|---|---|---|---|---|---|---|---|---|---|
| (A) test for pathogen prevalence theory | | | | | | | | | |
| historical pathogen epidemics | | | | | | | | | |
| β | −0.001 | −0.001 | 0.002 | 0.002 | 0.000 | 0.000 | −0.000 | 0.000 | 0.000 |
| *p*-value | 0.398 | 0.247 | 0.921 | 0.265 | 0.553 | 0.337 | 0.749 | 0.770 | 0.337 |
| N | 3956 | 3956 | 3956 | 3956 | 3956 | 3956 | 3956 | 3956 | 3956 |
| (B) test for general farming activities | | | | | | | | | |
| proportion of total cultivated land | | | | | | | | | |
| β | 0.027 | 0.005 | −0.059 | −0.123 | 0.038 | −0.037 | 0.020 | 0.048* | 0.003 |
| *p*-value | 0.386 | 0.987 | 0.312 | 0.929 | 0.789 | 0.348 | 0.123 | 0.081 | 0.659 |
| N | 4101 | 4101 | 4101 | 4101 | 4101 | 4101 | 4101 | 4101 | 4101 |

*** $p < 0.01$; ** $p < 0.05$; * $p < 0.10$.

that could explain this is the research on population density and 'life history' strategies [45]. Life-history research has found that people in densely populated places tend to shift their strategies from 'live fast, take chances' risky strategies to long-term investment strategies such as focusing time on fewer relationship partners and investing more in education. Areas with denser farming, in general, would have higher population densities and presumably more of the human institutions that go along with density, such as governments and schools.[3]

## 2.5. Why would rice select for these genes?

Overall, the data suggest that rice cultivation exerted selective pressures in favour of earlier reproduction and alcohol intolerance. This leaves the question of why rice farming might have selected for these specific traits. Although we cannot prove any particular theory from our data, we offer initial hypotheses based on the ecology of rice.

### 2.5.1. Rice and childbirth

Why might rice farming have selected for earlier childbirth? Three features of rice areas are consistent with this idea:

(1) Rice is far more labour-intensive than wheat and other dryland crops [3–5]. Researchers calculated how much labour a husband and wife would need to grow enough rice to eat and to barter for necessities like clothing and tools [46]. They concluded the labour demands were so high that a husband and wife would not be able to farm a large enough plot of rice to support the family if they relied on their labour alone. Children provided labour for farm families. Children were labour in farm families all over the world, but this may have been particularly critical for rice farmers. The idea that children could contribute to labour demands is consistent with one anthropologist's observation that rice farmers in China preferred to meet peak labour demands by enlisting family and extended family, rather than neighbours or wage labourers [46]. Thus, the environment for rice farming may have selected for people who had children at a younger age and thus had more offspring and labour in the family over a lifetime.

(2) Rice farming is more productive than crops like wheat and millet. Historically, paddy rice produced three to five times the output per acre as wheat [5]. The relative abundance of food might have encouraged earlier reproduction in rice-farming communities [47].

(3) There is also evidence that China's rice areas had shorter life expectancy than wheat areas historically [35]. There is evidence that people in rice areas had less-diverse diets (deficiency of certain nutrients such as vitamin B and iron) and denser populations in an era without sanitation infrastructure [35]. One historian argued that the high productivity of rice might have paradoxically created more catastrophic collapses [35]. Because rice was so productive, it would have supported a larger population, which would have incentivized people to turn more land into rice fields. China's impressive rice terraces—rice fields cut into steep mountains—hint at this conversion of even marginal land into rice land. Although this would have raised overall productivity, it would have made the local ecosystem more susceptible to periodic collapses from drought or crop diseases [35]. The shortened lifespan and risk of disaster could have plausibly increased pressure to have children at younger ages. Such a pattern has been observed in domesticated animals, which usually have shorter lifespans but have offspring at earlier ages and with higher frequency.

### 2.5.2. Rice and alcohol

Why might rice have selected for genotypes of aldehyde dehydrogenase deficiency? This gene leads to excessive accumulation of acetaldehyde in the liver and the alcohol-flushing reaction, sometimes called the 'Asian flush' [30]. Grain was a common ingredient used to make alcohol. Some researchers have theorized that rice farmers may have had earlier access to the excess grain used to make alcohols like rice wine as early as 9000 years ago [11,30]. 'Agriculture and the making of fermented beverages go together.

---

[3]We also perform a series of robustness checks by using proxies of historical rice farming, including the province-level percentage of cultivated land devoted to paddy rice in 1934, the province-level paddy rice yield per unit area (Mu or 亩 in Chinese) in the Qing dynasty, and the province-level total number of paddy rice varieties recorded by the administration of Ming (AD 1368–1644) and Qing (AD 1636–1912) dynasties. These additional results are consistent with original findings, and are available upon request from the corresponding author.

Most hunter–gatherer populations do not have the means, know-how, or resources' to make alcohol [11]. With more alcohol available, rice areas may have had more experience with the consequences, such as alcoholism, child neglect, altered innate immune modulation and tumour development [48]. Thus, rice areas may have been exposed to alcohol pressure for longer than other populations [11,30].

We offer initial theories for why rice might have selected for these genes, but these are early steps since there is still a lack of direct archaeological evidence (such as from ancient DNA). As historians and anthropologists uncover more about the history of rice farming, we can refine our understanding of how it might have selected for particular genes. Furthermore, these results await future replication in samples from China and other rice-farming populations around the world, from East Asia, to India, and West Africa.

## 2.6. Potential implications for modern society

PGSs are not destiny. The relationship between genes and behaviour can differ entirely in a different environment, such as the same country in an earlier era versus a later era. Thus, it is important to treat implications of genetic differences with caution.

However, these differences may be a starting point for researchers investigating regional differences in China or between China and other countries. One relatively straightforward implication is that the aldehyde dehydrogenase difference in southern China would lead to less problematic drinking and perhaps less drinking overall. This hypothesis would be fairly straightforward to test.

The genetic scores linked to earlier childbirth could lead to the prediction of higher fertility or earlier birth of the first child in rice-farming parts of China. However, birth rates have been falling across China in the last century, and modern forms of contraception may disrupt the link between gene-related fertility and actual childbirth. However, it is possible, for example, that rates of unintended pregnancies may be higher in rice-farming parts of China.

# 3. Discussion

In sum, genetic data from over 4000 people across China produced evidence that genes for earlier reproduction and alcohol flush response were more common among people from areas with more historical rice farming. Rice farming was negatively associated with PGSs for educational attainment, although this relationship became marginal after controlling for the history of herding.

The effect of rice remained robust after controlling for individual demographic characteristics, ethnic make-up, a range of regional characteristics and potential self-selection into rice farming. Moreover, the large sample size of counties substantially increases statistical power and allows for greater control over confounding factors in the analysis. The results of this study suggest that a major cultural transition in human history had small but detectable effects on genes.

Researchers used to believe that evolution worked so slowly that meaningful changes were unlikely to have happened in the last 10 000 years of human history. But more recently, researchers have concluded that 'evolutionary change typically occurs much faster than people used to think'. There is also evidence that human evolution actually *sped up* in the last 40 000 years [49]. If rice domestication selected for particular genes, it would fit with this emerging picture of relatively recent human evolution.

We should note several limitations in our data that point to possible future improvements. (i) The current study is based on a sample of 4101 observations, which may lack statistical power due to the small sample size. (ii) The GWAS summary statistics used to construct the PGSs in this study were mostly based on samples of European ancestry, which may lead to a Euro-centric bias and limit the predictive power constructed PGSs [14].[4] (iii) Identifying regional ancestry through the place of birth is not perfect. This method may misidentify people whose recent ancestors moved large distances. (iv) We analysed genetic differences but not phenotypes or actual behaviour. Genetic propensities are not destiny. (v) We do not have DNA samples from historical periods (e.g. ancient DNA). If future researchers gain access to historical DNA samples, this will allow for a directly test or completely rule out of the reverse causality issue.

It is worth remembering that environment is not destiny, either. It would be overly simplistic to expect that exact same pattern of results everywhere people grow rice. There is ample evidence that

---

[4]In a recent study, Duncan *et al*. [50] demonstrated that the polygenic score performance may be reasonably reliable in East Asian samples (95%) relative to European samples (100%). We expect the generalizability of polygenic scores to non-European ancestry populations to be an active area of methods development in the near future.

the same type of environment does not always lead to the same culture. As one small example, how farmers dealt with peak labour demands in rice differed across cultures. While Chinese farmers preferred to trade labour with family members, West African rice farmers sometimes relied on groups of youths, who would move from farm to farm. Rice presents common challenges, but cultures' solutions to those challenges (and the genetic selection pressures that come along) may differ.

Finally, the finding of rice–wheat genetic differences presents a hint about a puzzle of modernization. As fewer and fewer people are farming in China, how is it that rice–wheat differences persist in modern China? Studies have found rice–wheat differences among people who do not farm [8,9]. Genetic differences present one possible mechanism—but surely not the only mechanism—through which historical differences in subsistence style live on in the present day.

# 4. Methods

## 4.1. Data and research design

Through WeGene, we conducted an online survey with its customer base. WeGene is a leading private genetic testing company based in Shenzhen, which provides direct-to-consumer genetic testing and personalized healthcare information. After providing informed consent, approximately 4700 participants took our online survey between July 2018 and October 2019. The survey collected information on participants' demographic and socioeconomic characteristics such as gender, birth year, birthplace and parental birthplaces. Excluding individuals who did not finish enough questions or were under the age of 16 at the time of the survey yielded a total of 4101 observations from the original sample. An important feature about our dataset is that all respondents were genotyped on a WeGene custom genotyping array (Illumina). Imputation and quality control were performed using PLINK (1.90 Beta), SHAPEIT (v. 2.17) and IMPUTE2 (v. 2.3.1). For each individual, we obtained a total of 10 670 107 SNPs, which we then used to construct PGSs for a number of behavioural and psychological traits.

We built PGSs (also called 'polygenic risk scores', 'genetic risk scores' or 'genome-wide scores') for all 4101 individuals using effect estimates from recently published GWASs on height, BMI, depression, time discounting, age at first birth, educational attainments and risk preference [30–37]. We calculated each PGS as the sum of imputed allele $j$ dosages carried by a respondent $i$ ($SNP_{j,i}$) multiplied by the estimated effect size ($\beta_j$) reported by related GWAS, i.e. $PGS_i = \sum_{i=1}^{j} \beta_j SNP_{j.i}$. We then normalized all PGSs between 0 and 1.

Two exceptions are the *ADH1B* rs1229984 gene and *ALDH2* rs671 gene. *ADH1B* encodes alcohol dehydrogenase, and *ALDH2* encodes aldehyde dehydrogenase. *ADH1B* is solely determined by SNP rs1229984, and *ALDH2* is determined by SNP rs671 alone, rather than multiple SNPs. Therefore, we directly measure *ADH1B* rs1229984 and *ALDH2* rs671 as the number of effect alleles, resulting in three possible values (0, 1 and 2).

## 4.2. Statistical analysis

We estimate the following multivariate regression model using ordinary least squares (OLS), where $i$ denotes individuals and $j$ denotes county of birth

$$PGS_{ij} = \lambda(pt\_rice_{ij}) + \beta X_{ij} + \varepsilon_{ij}. \tag{4.1}$$

The dependent variable is respondent $i$ (born in county $j$), who has one of the nine PGSs or genotypes (i.e. height, BMI, *ADH1B*, *ALDH2*, depression, time discounting, age at first birth, educational attainments and risk preference); $pt\_rice_{ij}$ represents the proportion of rice farming in county $j$ that $i$ was born in, and $\lambda$ is the coefficient of primary interest; $X_{ij}$ contains a set of control variables, including individual characteristics, ancestral compositions and measures of county/province differences depending on the specific model described in the main text.

To mitigate the concern of rice farming endogeneity, we estimate two-stage least squares (2SLS) models using the following equations:

$$\text{first stage: } pt\_rice_{ij} = rice\_suitability_{ij}\delta_1 + rice\_suitability_{ij}^2\delta_2 + \mu X_{ij} + \xi_i \tag{4.2}$$

$$\text{second stage: } PGS_{ij} = \lambda(\widehat{pt\_rice_{ij}}) + X_{ij}\beta + \varepsilon_{ij}. \tag{4.3}$$

In the first stage (equation (4.2)), the endogenous variable of county-level rice farming percentage ($pt\_rice_{ij}$) is regressed on instruments of the environmental suitability for rice ($rice\_suitability_{ij}$) and its

quadratic form and the vector of control variables in $X$ as described earlier. In the second stage (equation (4.3)), different from in equation (4.1), each polygenic trait or genotype is then regressed on the fitted value of county-level rice farming (pt_rice) and control variables.

Ethics. This study was approved by the Ethics Committee of China Agricultural University (Approval Number: CAUHR-2020005). All participants provided signed informed consent before the data collection process.

Data accessibility. Data available from the Dryad Digital Repository: https://doi.org/10.5061/dryad.djh9w0w00 [51].

Authors' contributions. C.Z. and T.T. designed this study and drafted the manuscript. C.Z., J.Z., J.W., Y.L. and G.C. performed research and analysed the data. All the authors read and approved the final version of the manuscript.

Competing interests. We declare a competing interest.

Funding. This work was supported by the 2115 Talent Development Program of China Agricultural University; the Academy of Global Food Economics and Policy (AGFEP) and the Beijing Food Safety Policy & Strategy Research Base.

Acknowledgements. The authors are grateful to WeGene customers who answered the survey and participated in this research. The authors thank the staff of WeGene for their help in collecting the data. The authors also thank all the conference participants for the 23andMe Genome Research Day of 2019.

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
