## [Peer Review File · Royal Society Open Science]

Review History

RSOS-210382.R0 (Original submission)

Review form: Reviewer 1 (Petri Bäckerman)

Is the manuscript scientifically sound in its present form?

Yes

Are the interpretations and conclusions justified by the results?

Yes

Is the language acceptable?

Yes

Do you have any ethical concerns with this paper?

No

Have you any concerns about statistical analyses in this paper?

No

Recommendation?

Accept with minor revision (please list in comments)

Comments to the Author(s)

Comments

I find that the results that are presented in the paper are interesting and novel. Below I discuss issues that I think could help to improve the paper.

1. The data have information on 4101 individuals. Is the sample size sufficient to identify all the relevant patterns?
2. Climate changes over time. The regional pattern of rice farming in China could have been somewhat different in historical times (page 6). Does this have implications for the findings that are presented in the paper?
3. Is rice farming combined with other forms of farming? Have there been changes in this respect over time or are the relationships stable?
4. Is rice farming linked to other activities e.g., social patterns that could explain the findings that are presented in the paper?
5. Are there regions in which individuals abandoned rice farming?
6. Could the start of rice farming be explained by "genes" (i.e., endowments that are used in the paper to explain earlier reproduction and lower alcohol tolerance)? Is the exclusion restriction of the IV approach on page 16 that is used to address the issue plausible? Sargan test statistics is not a strong test for the validity of IV approach.
7. The effect of genes on the outcomes of interest could be non-linear.
8. The genes can have multiple (unknown) effects on the outcomes of interest and complex interactions between genes and environment cannot be rule out. This may limit the conclusions that can be drawn from the analysis.
9. Why the better availability of alcohol (caused by rice farming) does not lead to higher tolerance for alcohol?
10. Do the results have implications for the modern society/economy in China?

Review form: Reviewer 2

Is the manuscript scientifically sound in its present form?

No

Are the interpretations and conclusions justified by the results?

No

Is the language acceptable?

Yes

Do you have any ethical concerns with this paper?

Yes

Have you any concerns about statistical analyses in this paper?

Yes

Recommendation?

Major revision is needed (please make suggestions in comments)

Comments to the Author(s)

Thank you for the invitation to review the manuscript by Zhu et al. „Did Agriculture Shape Human Beings? Evidence of Genetic Adaptation to Rice Farming from China“. I think the

formulated hypothesis is interesting and the results are potentially relevant. However, some major aspects should be considered in advance to publication.

- I consider the title is misleading, since genetic adaptation is not directly/properly assessed in the study. The authors should just describe what they did, for example: Relationship between rice farming and polygenic risk scores potentially linked to agriculture in China“.

- Also the abstract should be reformulated, as the authors did not check variant selection. The introductory part of the abstract is quite long, but the investigated traits are not listed. The findings of the study are not described, but only their potential interpretation. Please note that adaptation, selection, selective pressure are all very specific terms in genetics.

- Ethnic confounding is a major issue in this study, and the authors should very carefully address it. I do not consider this falls within any “Alternative theory and Individual Control Variables”. Ethnicity should be included in all the statistical analyses conducted.

Were the 42 considered ancestral compositions (1) categorical or (2) the percentages of genetic ancestry estimated using ADMIXTURE? How many genetic principal components are typically considered in association studies of individuals of Chinese ancestry?

- Results should be adjusted for multiple testing and exact probability values should be provided (for example, 0.001 instead of ***).

- The transferability of PRSs developed for Europeans to other populations is controversial. The authors should assess the prediction performance of the used PRSs in the investigated study population, individual information on height, BMI, education attainment and, particularly important due to the results, age at first birth should be available. The general paragraph “recent studies suggest that the results can well apply to East Asian (e.g., Chinese) populations. For example, Duncan and colleagues (34) demonstrated that the polygenic score performance is reasonably reliable in East Asian samples (95%) relative to European samples (100%).” is not enough.

- Please replace Fig. 2 by scatter plots with the same axes and (non)-linear regression lines if needed.

- For testing reverse causality, I would suggest simply to interchange the response and explanatory variables in the OLS and 2SLS analyses.

Decision letter (RSOS-210382.R0)

Dear Dr Zhu

The Editors assigned to your paper RSOS-210382 "Did Agriculture Shape Human Beings? Evidence of Genetic Adaptation to Rice Farming from China" have now received comments from reviewers and would like you to revise the paper in accordance with the reviewer comments and any comments from the Editors. Please note this decision does not guarantee eventual acceptance.

Both reviewers find the work interesting, but both of them raise a large number of substantive points and criticisms that will require careful consideration and appropriate revision. We invite you to respond to the comments supplied below and revise your manuscript. Below the referees' and Editors' comments (where applicable) we provide additional requirements. Final acceptance of your manuscript is dependent on these requirements being met. We provide guidance below to help you prepare your revision.

Please submit your revised manuscript and required files (see below) no later than 21 days from today's (ie 21-May-2021) date. Note: the ScholarOne system will 'lock' if submission of the revision is attempted 21 or more days after the deadline. If you do not think you will be able to meet this deadline please contact the editorial office immediately.

on behalf of Professor Andrés Ruiz-Linares (Associate Editor) and Steve Brown (Subject Editor)
openscience@royalsociety.org

Associate Editor Comments to Author (Professor Andrés Ruiz-Linares):

Please revise the paper taking into account all the comments of the two reviewers.

Reviewer comments to Author:

Reviewer: 1
Comments to the Author(s)
Comments

I find that the results that are presented in the paper are interesting and novel. Below I discuss issues that I think could help to improve the paper.

1. The data have information on 4101 individuals. Is the sample size sufficient to identify all the relevant patterns?

2. Climate changes over time. The regional pattern of rice farming in China could have been somewhat different in historical times (page 6). Does this have implications for the findings that are presented in the paper?
3. Is rice farming combined with other forms of farming? Have there been changes in this respect over time or are the relationships stable?
4. Is rice farming linked to other activities e.g., social patterns that could explain the findings that are presented in the paper?
5. Are there regions in which individuals abandoned rice farming?
6. Could the start of rice farming be explained by “genes” (i.e., endowments that are used in the paper to explain earlier reproduction and lower alcohol tolerance)? Is the exclusion restriction of the IV approach on page 16 that is used to address the issue plausible? Sargan test statistics is not a strong test for the validity of IV approach.
7. The effect of genes on the outcomes of interest could be non-linear.
8. The genes can have multiple (unknown) effects on the outcomes of interest and complex interactions between genes and environment cannot be rule out. This may limit the conclusions that can be drawn from the analysis.
9. Why the better availability of alcohol (caused by rice farming) does not lead to higher tolerance for alcohol?
10. Do the results have implications for the modern society/economy in China?

Reviewer: 2

Comments to the Author(s)

Thank you for the invitation to review the manuscript by Zhu et al. „Did Agriculture Shape Human Beings? Evidence of Genetic Adaptation to Rice Farming from China“. I think the formulated hypothesis is interesting and the results are potentially relevant. However, some major aspects should be considered in advance to publication.

- I consider the title is misleading, since genetic adaptation is not directly/properly assessed in the study. The authors should just describe what they did, for example: Relationship between rice farming and polygenic risk scores potentially linked to agriculture in China“.

- Also the abstract should be reformulated, as the authors did not check variant selection. The introductory part of the abstract is quite long, but the investigated traits are not listed. The findings of the study are not described, but only their potential interpretation. Please note that adaptation, selection, selective pressure are all very specific terms in genetics.

- Ethnic confounding is a major issue in this study, and the authors should very carefully address it. I do not consider this falls within any “Alternative theory and Individual Control Variables“. Ethnicity should be included in all the statistical analyses conducted. Were the 42 considered ancestral compositions (1) categorical or (2) the percentages of genetic ancestry estimated using ADMIXTURE? How many genetic principal components are typically considered in association studies of individuals of Chinese ancestry?

- Results should be adjusted for multiple testing and exact probability values should be provided (for example, 0.001 instead of ***).

- The transferability of PRSs developed for Europeans to other populations is controversial. The authors should assess the prediction performance of the used PRSs in the investigated study population, individual information on height, BMI, education attainment and, particularly important due to the results, age at first birth should be available. The general paragraph “recent studies suggest that the results can well apply to East Asian (e.g., Chinese) populations. For example, Duncan and colleagues (34) demonstrated that the polygenic score performance is

reasonably reliable in East Asian samples (95%) relative to European samples (100%).” is not enough.

- Please replace Fig. 2 by scatter plots with the same axes and (non)-linear regression lines if needed.

- For testing reverse causality, I would suggest simply to interchange the response and explanatory variables in the OLS and 2SLS analyses.

===PREPARING YOUR MANUSCRIPT===

===PREPARING YOUR REVISION IN SCHOLARONE===

Please ensure that you include a summary of your paper at Step 2 'Type, Title, & Abstract'. This should be no more than 100 words to explain to a non-scientific audience the key findings of your

research. This will be included in a weekly highlights email circulated by the Royal Society press office to national UK, international, and scientific news outlets to promote your work.

Author's Response to Decision Letter for (RSOS-210382.R0)

See Appendix A.

RSOS-210382.R1 (Revision)

Review form: Reviewer 1 (Petri Bäckerman)

Is the manuscript scientifically sound in its present form?

Yes

Are the interpretations and conclusions justified by the results?

Yes

Is the language acceptable?

Yes

Do you have any ethical concerns with this paper?

No

Have you any concerns about statistical analyses in this paper?

No

Recommendation?

Accept as is

Comments to the Author(s)

I am happy with the revised version of the paper.

Decision letter (RSOS-210382.R1)

Dear Dr Zhu,

It is a pleasure to accept your manuscript entitled "Evidence of Genetic Differences Explained by Rice Agriculture in China" in its current form for publication in Royal Society Open Science. The comments from the Editor and reviewers who reviewed your manuscript are included at the foot of this letter.

Please see the Royal Society Publishing guidance on how you may share your accepted author manuscript at <https://royalsociety.org/journals/ethics-policies/media-embargo/>. After

publication, some additional ways to effectively promote your article can also be found here <https://royalsociety.org/blog/2020/07/promoting-your-latest-paper-and-tracking-your-results/>.

on behalf of Professor Andrés Ruiz-Linares (Associate Editor) and Steve Brown (Subject Editor)
openscience@royalsociety.org

Associate Editor Comments to Author (Professor Andrés Ruiz-Linares):

Many thanks for the extensive revision of the paper. I would, however, suggest you consider aligning the title of the paper more closely to the one suggested by reviewer #2, which I find more informative than the current one: "Relationship between rice farming and polygenic risk scores potentially linked to agriculture in China"

Reviewer comments to Author:

Reviewer: 1

Comments to the Author(s)

I am happy with the revised version of the paper.

Follow Royal Society Publishing on Twitter: [@RSocPublishing](https://twitter.com/RSocPublishing)

Appendix A

Journal: Royal Society Open Science

Manuscript ID: RSOS-210382

Response to Reviewers' comments on:

Did Agriculture Shape Human Beings? Evidence of Genetic Adaptation to Rice

Farming from China

Response to Reviewers' Comments

Before addressing reviewers' comments, we would like to thank the reviewers and editor for the time and energy devoted to reviewing our manuscript. We greatly appreciate the constructive suggestions for improving our paper. The manuscript has been carefully revised based on all reviewers' comments. Please find our point-by-point responses below.

Response to Reviewer #1's Comments

Q1. I find that the results that are presented in the paper are interesting and novel. Below

I discuss issues that I think could help to improve the paper.

The data have information on 4101 individuals. Is the sample size sufficient to identify all the relevant patterns?

Response: Thanks for pointing this out. We first calculate the minimum sample size to test the entire full model (Table 3, Model 6, total number of predictors = 50) as suggested by Green (1991), resulting in $N_{\text{minimal}} = 50 + 8 * 50 = 450$, which is smaller than our sample ($N=4,101$). We then calculate the sample sizes needed in more favourable scenario with 0.05 type I error and 0.8 power as suggested by (Hsieh et al., 1998), and the $N_{\text{favorable}}$ ranges between 1,995 to 19,785 depending on different outcome variables. Therefore, estimation results of some of the outcome variables can indeed suffer from a lack of statistical power due to the small sample size (Green, 1991). We add this limitation in the revision (on page 19).

Q2. Climate changes over time. The regional pattern of rice farming in China could have been somewhat different in historical times (page 6). Does this have implications for the findings that are presented in the paper?

Response: That is a good question. In the manuscript, the complete recorded for *county-level* rice farming data we could find is around the year 2000. To check this against historical records, we visited the National Library of China and found several sources of earlier *province-level* rice farming data:

- (1) The province-level percentage of cultivated land devoted to paddy rice in 1914-1918 and separately in 1934.
- (2) The province-level paddy rice yield per unit area ($mu, \text{畝}$) in the Qing dynasty.
- (3) The province-level total number of paddy rice varieties recorded by the administration of Ming (A.D. 1368-1644) and Qing (A.D. 1636-1912) dynasties.
- (4) The oldest age of rice relics per province for 21 provinces (dating back as far as 11,000 years).

Note that the Ming, Qing, and archaeological data is about rice, although it is not quite the same as the percentage of farmland devoted to rice. However, it is likely that areas that were growing more rice historically would also have older archaeological evidence of rice.

Stability of Rice Statistics Over History

As shown in Figure R1, the age of rice relics was strongly correlated with 1996 province-level rice data, $r(19) = .81, p < .001$. Results were similar for correlating 1914-1918 data and the 1996 data: $r(22) = 0.95, p < .001$. This is particularly impressive given the differences in measurement methods between 1914 and 1996. This picture is consistent with data from the United Nations Food and Agriculture Organization on the environmental suitability for rice. This data incorporates many physical factors, such as rainfall, sunlight, soil quality, and slope. This model calculates the suitability for rice, regardless of whether people are farming rice there or not. Although there have been climatic changes over time, this dataset (1) includes variables that are mostly stable over time (such as the slope of the land) and (2) other variables that do change, but are moderately stable over time (such as southern China being warmer than northern China). Provinces' environmental suitability scores strongly correlated with actual rice farming $r(29) = .87, p < .001$.

In sum, climate data, data from 100 years ago, and data from 11,000 years ago are

highly correlated with relatively modern statistics. This is not to say that there is **zero** variation over time. However, the broad outlines of rice and wheat seem to have been largely stable over time.

Figure R1. Correlations Between Present and Historical Rice Farming Measures

(a)

(b)

Modelling Historical Rice Statistics and Genetic Differences

As a next step, we ran models with historical rice data predicting differences in the genetic data. We used rice yield per area and the total number of rice varieties as proxies to the development of rice agriculture in the local area. We performed a series of checks by replacing the original county-level rice farming variable with these three historical rice farming variables.

Table R1 reports parameter estimates from separate multivariate regressions of each polygenic trait on the provincial percentage of paddy rice in 1934 (Model 1), provincial rice yield in Qing dynasty (Model 2), and the provincial total number of rice varieties in Ming and Qing dynasties (Model 3). All models are adjusted for age, growing-up in town and city, temperature, distance from the coast, GDP, contact with herding cultures, history of rebellions, ethnic ancestry, latitude and longitude as in the full model (Table 3, Model 6 in the revised manuscript).

As shown in Table R1, the province-level rice yield per unit area in the Qing dynasty and the total number of rice varieties in Ming and Qing dynasties are significantly associated with *ALDH2* rs671 and the polygenic score of age at first birth at 5% level (Model 2 and 3, column 4 and 7). The province-level percentage of paddy rice in 1934 also significantly predicts *ALDH2* rs671 (Model 1, column 4). Although its association with the polygenic score of age at first birth is marginally significant (Model 1, column 7), the sign and magnitude of all parameters remain consistent. Overall, these results using additional measures of historical rice farming data provide generally consistent results compared with using the county-level rice farming variable as in the original manuscript. We add a description of these results on page 16 of the revised manuscript.

Table R1. Additional Estimation Results

Outcome Variables: Polygenetic Traits	(1)	(2)	(3)	(4)	(5)	(6)	(7)	(8)	(9)
Key Dependent Variables	Height	BMI	ADH	ALDH	Depression	Time discounting	Age at first birth	Educational attainments	Risk preference
Model 1: Provincial % paddy rice field in 1934	-0.0151 (0.0150)	0.0085 (0.0156)	0.0048 (0.0732)	0.2933*** (0.0566)	-0.0070 (0.0135)	0.0236 (0.0187)	-0.0254* (0.0143)	-0.0171* (0.0090)	0.0099 (0.0135)
Model 2: Provincial paddy rice yield per unit area in Qing dynasty	-0.0004 (0.0021)	-0.0018 (0.0022)	-0.0148 (0.0102)	0.0457*** (0.0079)	-0.0013 (0.0019)	0.0013 (0.0026)	-0.0057*** (0.0020)	-0.0026** (0.0013)	0.0011 (0.0019)
Model 3: Provincial total number of paddy rice varieties in Ming and Qing dynasties	0.0001 (0.0003)	-0.0006* (0.0003)	-0.0011 (0.0015)	0.0046*** (0.0012)	-0.0004 (0.0003)	0.0000 (0.0004)	-0.0007** (0.0003)	-0.0003 (0.0002)	0.0001 (0.0003)
Model 4: County-level % paddy rice field	0.0083 (0.0193)	-0.0070 (0.0200)	-0.1687* (0.0939)	0.2455*** (0.0727)	0.0033 (0.0173)	0.0181 (0.0241)	-0.0460** (0.0184)	-0.0172 (0.0116)	0.0173 (0.0174)
Provincial total number of Confucian temples in Qing dynasty	-0.0006 (0.0013)	0.0042*** (0.0013)	0.0070 (0.0063)	0.0052 (0.0049)	0.0006 (0.0012)	0.0008 (0.0016)	-0.0008 (0.0012)	-0.0003 (0.0008)	0.0009 (0.0012)
N	4,101	4,101	4,101	4,101	4,101	4,101	4,101	4,101	4,101

Notes: ***, **, and * indicate statistical significance at the 1%, 5%, and 10% levels, respectively. Other covariates include age, growing-up in town and city, temperature, distance from the coast, GDP, contact with herding cultures, regional education, length of rivers, history of rebellions, ethnic ancestry, latitude and longitude.

Q3. Is rice farming combined with other forms of farming? Have there been changes in this respect over time or are the relationships stable?

Response: This is an interesting question. Over the long course of history (say, comparing 2,000 years ago to 100 years ago), cropping patterns have generally intensified (Elvin, 2008). For example, in the last few hundred years, Guangdong in southern China has farmed two or even three rice crops a year. Around 1,000 AD, they were mostly farming one crop a year. Farming intensity has increased as population density increased (providing more labour and a need for more calories), as technology has improved (for example, methods for irrigation), and as people have developed varieties of rice that ripen quicker.

What would this mean for genetic differences? One implication is that selection pressures of rice farming may have even intensified into the last few hundred years. The selection pressure of rice farming may not be just a story of thousands of years ago. For example, the intensity of rice farming and the selection pressures it engendered may have been stronger under triple cropping in the late 1800s than under single cropping in 1,000 AD.

The question also gets at the idea of interspersing other types of crops. This has also increased over time. For example, across central China, a common setup is to farm rice in the summer and wheat in the winter (see map below in Figure R2).

Figure R2. Map of Major Crops in China

The farming statistics we are using in the current study actually tap into this variation. Farming statistics will "double count" the same plot of land if it grows different crops in two seasons. For example, if a farmer grows wheat in the winter and paddy rice in the summer, the statistic would count as 50% land devoted to paddy rice.

Q4. Is rice farming linked to other activities e.g., social patterns that could explain the findings that are presented in the paper?

Response: This is a great question. This also plays into the question above about increasing intensity over time. as the reviewer hints at, we postulate that different farm crops were linked to social patterns that may have exerted selection pressures. One potential mechanism is the clan system. One researcher has argued that the development of agriculture (and rice farming in particular) helped to lead to the family clan system (Cohen, 2011). Such kinship-based Chinese clan culture deeply rooted in the Confucian ethics (Han, 2012; Slote and De Vos, 1998).

To follow up on this potential mechanism, we collected the provincial total number of Confucian temples in the Qing dynasty (as a proxy of the degree of Confucian influence and clan culture, following Kung and Ma (2014). Table R1, Model 4 reports parameter estimates from separate multivariate regressions of each polygenic trait on the county-

level percentage of paddy rice and the province-level total number of Confucian temples in the Qing dynasty. From the results, the county-level rice variable still significantly predicts *ALDH2* rs671 (column 4) and the polygenic score of age at first birth (column 7) at the 5% level, even after adjusting for the number of local Confucian temples in the Qing dynasty. Meanwhile, the historical number of Confucian temples is positively associated with a higher polygenic score of BMI (column 2), which is interesting but beyond the scope of the current study and may be valuable in future studies. Overall, estimation results controlling for local Confucian influence and clan culture are generally consistent with main findings in the original manuscript.

Q5. Are there regions in which individuals abandoned rice farming?

Response: That's an interesting question! Other than temporary shocks (such as warfare), we believe it was rare for regions to abandon rice farming. Instead, rice farming generally expanded over the course of history. Rice farming was first widespread around the Yangtze River (such as Hubei and Shanghai) and then spread further south to places like Guangdong (Elvin, 2008).

In very recent times (after the year 2000), rice has spread to far northeastern China (the *Dongbei*). These regions can now farm rice because modern diesel pumps have given them access to underground water to flood the fields. We think this modern spread is unlikely to have influenced genetics in the northeast because (1) it is very recent, (2) rice is still less than 20% of farmland in the northeast, and (3) farmers are using modern technologies like diesel pumps and tractors, which weaken the requirements for labour and interdependence.

Except for this modern expansion, the overall picture is that the boundaries of rice have been fairly stable for hundreds of years. The statistics we cite for Q2 above speak to the stability of rice farming over time.

Q6. Could the start of rice farming be explained by "genes" (i.e., endowments that are used in the paper to explain earlier reproduction and lower alcohol tolerance)? Is the exclusion restriction of the IV approach on page 16 that is used to address the issue plausible? Sargan test statistics is not a strong test for the validity of IV approach.

Response: This is a great question. We agree with the reviewer that it is conceivable that people who started growing rice 10,000 years ago may not be randomly selected from the whole population. In other words, ancient populations who grew rice

(southern China) and millet (northern China) may have been genetically different to start with.

Unfortunately, there is currently no paleoanthropological evidence or data (e.g., ancient DNA) that can be used to directly test such potential differences. Although this sort of "silver bullet" is lacking, we take several strategies to address this possibility:

(1) To minimize the concern that there were genetic variations inherited from ancestral populations, we included a total of 42 individual ancestral composition variables based on each respondent's genetic data as control variables. In our data, individual ancestry composition is estimated by using the ADMIXTURE program, developed by the Department of Human Genetics, University of California Los Angeles (Alexander and Lange, 2011). Table R2 reports the descriptive information of the top ten ancestries estimated in our sample. This includes northern versus southern Han populations, which would be the mostly likely candidate for inherited ancestral characteristics. The fact that the associations with rice hold even after controlling for these groupings suggests that rice differences are independent from group differences at the start.

Table R2. Descriptive Statistics of Top 10 Ancestries

Ancestry/Population	Mean	Std.Dev.	Min	Max
Northern Han	0.5530	0.2963	0.0000	0.9996
Southern Han	0.2579	0.2645	0.0000	0.9996
Mongolian	0.0604	0.1124	0.0000	0.7774
Naxi/Yi	0.0292	0.0610	0.0000	0.9996
Japanese	0.0202	0.0369	0.0000	0.2173
Gaoshan	0.0084	0.0171	0.0000	0.1163
Korean	0.0075	0.0105	0.0000	0.0575
Dai	0.0070	0.0203	0.0000	0.1696
She	0.0056	0.0096	0.0000	0.0540
Kinh	0.0054	0.0147	0.0000	0.1260

Source: author's calculation.

(2) We use an instrumental variable (environmental suitability for rice) to test for differences in a variable that is outside of human control (following Talhelm et al., 2014). The finding that environmental suitability for rice significantly predicts these genetic differences suggests that it is unlikely to be a case of reverse causality.

(3) Following the strategy of two prior studies (Conley et al., 2012; van Kippersluis and Rietveld, 2018), we tested whether our parameter estimates remain significant even with substantial relaxation of the assumption of strict exogeneity of the instrumental variables. Specifically, we use the "plausibly exogenous approach" built upon the inference strategy of plausibly exogenous instruments (Conley et al., 2012). This strategy can produce unbiased estimators when there are violations of the strict exclusion restriction. It can also be used as a sensitivity test when the exclusion restriction is uncertain or unknown (van Kippersluis and Rietveld, 2018).

Table R3 below shows the results. To make it easier to compare, we also reproduced the main OLS and 2SLS results from the main text. The estimated coefficients for paddy rice using the plausibly exogenous approach (Row 3) are consistent with the key 2SLS findings (Row 2). This suggests that violations to the exclusion restriction assumption are not a critical issue in the data. We add a description of these results on page 15 in the revised manuscript.

Table R3. OLS, 2SLS, and Plausibly Exogenous Results

Outcome variables: polygenetic scores (or genotypes)	(1)	(2)	(3)	(4)	(5)	(6)	(7)	(8)	(9)
	Height	BMI	ADH	ALDH	Depression	Time discounting	Age at first birth	Educational attainments	Risk preference
OLS	0.003 (0.022)	0.009 (0.023)	-0.136 (0.106)	0.251*** (0.081)	0.005 (0.020)	0.038 (0.027)	-0.046** (0.021)	-0.013 (0.013)	0.028 (0.020)
2SLS	-0.003 (0.013)	-0.002 (0.013)	0.020 (0.062)	0.182*** (0.046)	-0.003 (0.012)	0.010 (0.016)	-0.029** (0.011)	-0.015 (0.012)	-0.002 (0.011)
Plausibly Exogeneous	-0.0082 (0.0247)	-0.0229 (0.0258)	-0.0898 (0.1956)	0.3419*** (0.0924)	0.0087 (0.0222)	0.0098 (0.0308)	-0.0483** (0.0236)	-0.0208 (0.0148)	0.0306 (0.0223)
N	4,101	4,101	4,101	4,101	4,101	4,101	4,101	4,101	4,101

Notes: ***, **, and * indicate statistical significance at the 1%, 5%, and 10% levels, respectively. Other covariates include age, growing-up in town and city, ethnic ancestry, temperature, distance from coast, GDP, contact with herding cultures, history of rebellions, latitude and longitude.

Q7. The effect of genes on the outcomes of interest could be non-linear.

Response: Thanks for pointing this out. Due to unclarity in some parts of our original manuscript, the reviewer might have misunderstood our research question. We seek to examine the impact of paddy rice farming on genes (i.e., polygenic scores and genotypes), instead of the effect of genes on other outcomes of interest. In the revised draft, we have clarified to avoid further confusion.

Q8. The genes can have multiple (unknown) effects on the outcomes of interest and complex interactions between genes and environment cannot be rule out. This may limit the conclusions that can be drawn from the analysis.

Response: This is a fair point. It is true that genes can have multiple effects on an outcome variable through different pathways and interactions with environmental factors. It is worth pointing out that our outcome measure is genes, not behavior. Thus, complex interactions are not so much a problem for interpreting the relationships we analyze (between rice and genes, each of which is measured fairly precisely). However, as the reviewer points out, it is important to make it clear to readers that there can be variability in the relationship between genes and behaviour. We've revised the paper to make it clearer that our outcomes are genes and not behaviours.

Q9. Why the better availability of alcohol (caused by rice farming) does not lead to higher tolerance for alcohol?

Response: That is a good question! It seems like one could predict the opposite as well. For this, we rely on prior research and theorizing. Here we rely on prior theorizing (such as Henrich, 2015). Researchers have theorized that the selection of aldehyde dehydrogenase deficiency by rice farming (and earlier availability of alcohol) may be a response to avoid the consequences of excessive drinking and alcoholism, including child neglect, altered innate immune modulation, and tumour development.

However, it is worth pointing out for readers that this explanation is an initial theory (like many proposed explanations for genetic adaptation). These theories require further investigation, such as phylogenetic studies that compare samples of dozens of different ethnicities and attempt to trace when behaviours arose or were dropped (there is a good example in a study that analyzed when different historical populations started practising dairying: Holden and Mace, 2009). In the revised draft, we have clarified on page 18 to avoid further confusion.

Q10. Do the results have implications for the modern society/economy in China?

Response: This is a good question! We feel like we have to thread the needle between (1) discussing interesting potential implications and (2) avoiding speculating from genetic data to behaviour. But, as long as we write carefully and explicitly, we think it is possible to write productively about potential implications for modern society. With that in mind, we added a section titled "Potential Implications for Modern Society" on page 18.

References:

Alexander, D.H. and Lange, K., 2011. Enhancements to the ADMIXTURE algorithm for individual ancestry estimation. *BMC bioinformatics*, 12(1), p.246.

Cohen, D. J. (2011). The beginnings of agriculture in China: A multiregional view. *Current Anthropology*, 52(S4), S273-S293.

Conley, T. G., Hansen, C. B., & Rossi, P. E. (2012). Plausibly exogenous. *Review of Economics and Statistics*, 94(1), 260-272.

Elvin, M. (2008). *The retreat of the elephants: An environmental history of China*. Yale University Press.

Green, S. B. (1991). How many subjects does it take to do a regression analysis. *Multivariate Behavioral Research*, 26(3), 499-510.

Han, Q. (2012). The ties that bind: An overview of traditional Chinese family ethics. *Quarterly Journal of Chinese Studies*, 1(1), 85-99.

Henrich, J. (2015). *The secret of our success: How culture is driving human evolution, domesticating our species, and making us smarter*. Princeton University Press.

Holden, C., & Mace, R. (2009). Phylogenetic analysis of the evolution of lactose digestion in adults. *Human Biology*, 81(5/6), 597–620.

Hsieh, F. Y., Bloch, D. A., & Larsen, M. D. (1998). A simple method of sample size calculation for linear and logistic regression. *Statistics in Medicine*, 17(14), 1623-1634.

Kung, J. K. S., & Ma, C. (2014). Can cultural norms reduce conflicts? Confucianism and peasant rebellions in Qing China. *Journal of Development Economics*, 111, 132-149.

Lu, H., Zhang, J., Liu, K. B., Wu, N., Li, Y., Zhou, K., ... & Li, Q. (2009). Earliest domestication of common millet (*Panicum miliaceum*) in East Asia extended to 10,000 years ago. *Proceedings of the National Academy of Sciences*, 106(18), 7367-7372.

McGovern, P. E., Zhang, J., Tang, J., Zhang, Z., Hall, G. R., Moreau, R. A., ... & Wang, C. (2004). Fermented beverages of pre-and proto-historic China. *Proceedings of the National Academy of Sciences*, 101(51), 17593-17598.

Slote, W. H., & De Vos, G. A. (Eds.). (1998). *Confucianism and the family*. Suny Press.

Talhelm, T., Zhang, X., Oishi, S., Shimin, C., Duan, D., Lan, X., & Kitayama, S. (2014). Large-scale psychological differences within China explained by rice versus wheat agriculture. *Science*, 344(6184), 603-608.

Talhelm, T., & Oishi, S. (2018). How Rice Farming Shaped Culture in Southern China. *Socio-economic Environment and Human Psychology: Social, Ecological, and Cultural Perspectives*, 53.

Talhelm, T., Zhang, X., & Oishi, S. (2018). Moving chairs in Starbucks: Observational studies find rice-wheat cultural differences in daily life in China. *Science advances*, 4(4), eaap8469.

Talhelm, T., & English, A. S. (2020). Historically rice-farming societies have tighter social norms in China and worldwide. *Proceedings of the National Academy of Sciences*, 117(33), 19816-19824.

van Kippersluis, H., & Rietveld, C. A. (2018). Beyond plausibly exogenous. *The Econometrics Journal*, 21(3), 316-331.

Yang, X., Wan, Z., Perry, L., Lu, H., Wang, Q., Zhao, C., ... & Ge, Q. (2012). Early

millet use in northern China. Proceedings of the National Academy of Sciences, 109(10), 3726-3730.

Response to Reviewer #2's Comments

Q1. Thank you for the invitation to review the manuscript by Zhu et al., "Did Agriculture Shape Human Beings? Evidence of Genetic Adaptation to Rice Farming from China". I think the formulated hypothesis is interesting and the results are potentially relevant. However, some major aspects should be considered in advance to publication.

I consider the title is misleading, since genetic adaptation is not directly/properly assessed in the study. The authors should just describe what they did, for example: "Relationship between rice farming and polygenic risk scores potentially linked to agriculture in China".

Response: Thanks for the helpful suggestion. We have changed the title to "Evidence of Genetic Differences Explained by Rice Agriculture in China" to avoid further confusion.

Q2. Also the abstract should be reformulated, as the authors did not check variant selection. The introductory part of the abstract is quite long, but the investigated traits are not listed. The findings of the study are not described, but only their potential interpretation. Please note that adaptation, selection, selective pressure are all very specific terms in genetics.

Response: Thanks for the helpful suggestion. We have re-written the abstract as the reviewer suggested:

"Following domestication in the lower Yangtze River valley 9,400 years ago, rice farming spread throughout China and changed lifestyle patterns among Neolithic populations. Here we report evidence that the advent of rice domestication and cultivation may have shaped humans not only culturally but also genetically. Leveraging recent findings from molecular genetics, we construct a number of

polygenic scores of behavioural traits and examine their associations with rice cultivation based on a sample of 4,101 individuals recently collected from Mainland China. A total of nine polygenic traits and genotypes are investigated in this study, including polygenic scores of height, body mass index, depression, time discounting, reproduction, educational attainment, risk preference, ADH1B rs1229984 and ALDH2 rs671. Two-stage least squares estimates of the county-level percentage of cultivated land devoted to paddy rice on the polygenic score of age at first birth ($b = -0.029$, $p = 0.021$) and ALDH2 rs671 ($b = 0.182$, $p < 0.001$) are both statistically significant and robust to a wide range of potential confounds and alternative explanations. These findings imply that rice farming may influence human evolution in relatively recent human history."

Q3. Ethnic confounding is a major issue in this study, and the authors should very carefully address it. I do not consider this falls within any "Alternative theory and Individual Control Variables". Ethnicity should be included in all the statistical analyses conducted. Were the 42 considered ancestral compositions (1) categorical or (2) the percentages of genetic ancestry estimated using ADMIXTURE? How many genetic principal components are typically considered in association studies of individuals of Chinese ancestry?

Response: Thanks for pointing this out. As the reviewer suggested, we have included ancestral controls in all models and updated Table 3 in the revised manuscript. The 42 ancestral compositions are percentages of genetic ancestry estimated using ADMIXTURE. Table R2 reports the descriptive information of the top ten ancestries estimated in our sample. The Principal Component Analysis (PCA) is essentially a statistical strategy to reduce the complexity and overcome the curse of dimensionality of datasets, with a cost of loss of information (albeit minimal; Peterson et al., 2017), and there are typically 5-10 genetic principal components used in studies of Chinese populations (Chen et al., 2009; Li et al., 2021). Theoretically, when all ancestral compositions are included as in our analysis, there should be no loss of information (Gomez and Moens, 2012). We thus remain with the adjustment of population structure by using the 42 ancestral composition variables.

Q4. Results should be adjusted for multiple testing and exact probability values should

be provided (for example, 0.001 instead of ***).

Response: Thanks for pointing this out. The fact that we consider nine different outcomes raises concerns about multiple hypothesis testing (MHT) and false positives. Fail to account for MHT can lead to a substantial risk of false discoveries in the econometric analysis (Romano and Wolf, 2005). In order to address the issue, we implement a common p-value correction method of the D/AP procedure (Sankoh et al., 1997), and report both unadjusted and adjusted results after the estimation of main models. A major advantage of the D/AP procedure is that it can take into account the correlation structure among all nine outcome variables (Sankoh et al., 1997; Perneger, 1998). Specifically, we employ the following D/AP procedure: for correlated N outcome variables to be tested, the adjusted p -value is calculated as

$p_{1,adjusted} = 1 - (1 - p_1)^{M1}$, where $M1 = N^{1-r1}$, $r1 = (N-1)^{-1} \sum_{j \neq n}^N r_{jn}$, and r_{jn} is the correlation

coefficient between the j^{th} and n^{th} outcome variables, and so forth. Table R4 presents main OLS and 2SLS parameter estimates of paddy rice on outcome variables, deriving from both unadjusted and D/AP adjusted p-values for multiple comparisons. As shown in the table, after accounting for multiple testing by applying the D/AP correction, the 2SLS estimate of paddy rice on the polygenic score of age at first birth becomes marginally significant (p-value = 0.079). It is worth noting that the OLS and 2SLS estimates of paddy rice on *ALDH2* rs671 and OLS estimate of paddy rice on $PGS_{\text{age at first birth}}$ remain statistically significant using the adjusted p-values. Overall, these additional results after accounting for MHT are generally consistent with the main findings. We have added this discussion on page 11 in the revised manuscript. In addition, as the reviewer suggested, we also added the exact probability values and updated tables in the revised manuscript.

Table R4. Main Estimation Results After Accounting for Multiple Hypothesis Testing

	(1)	(2)	(3)	(4)	(5)	(6)	(7)	(8)	(9)
Outcome variables: polygenetic scores (or genotypes)	Height	BMI	ADH	ALDH	Depression	Time discounting	Age at first birth	Educational attainments	Risk preference
(A) Multivariate Regressions,									
full model:									
beta	0.003	0.009	-0.136	0.251	0.005	0.038	-0.046	-0.013	0.028
p-value (unadjusted)	0.894	0.704	0.198	0.002***	0.789	0.158	0.025**	0.331	0.157
p-value (D/AP adjusted)	1.000	0.984	0.554	0.003***	0.997	0.469	0.047**	0.759	0.466
(B) Two stage least squares (2SLS)									
beta	-0.003	-0.002	0.020	0.182	-0.003	0.010	-0.029	-0.015	-0.002
p-value (unadjusted)	0.741	0.373	0.108	< 0.001***	0.695	0.750	0.021**	0.161	0.169
p-value (D/AP adjusted)	0.995	0.797	0.029	0.001***	0.987	0.994	0.079*	0.463	0.494
First stage F statistic	84.740	84.740	84.740	84.740	84.740	84.740	84.740	84.740	84.740
p-value	0.000	0.000	0.000	0.000	0.000	0.000	0.000	0.000	0.000
Sargan statistic	2.183	2.445	2.372	2.103	3.087	2.535	2.185	1.805	2.218
p-value	0.140	0.118	0.124	0.147	0.079	0.111	0.139	0.179	0.136
N	4,101	4,101	4,101	4,101	4,101	4,101	4,101	4,101	4,101

Notes: ***, **, and * indicate statistical significance at the 1%, 5%, and 10% levels, respectively. Other covariates include age, growing-up in town and city, ethnic ancestry, temperature, distance from the coast, GDP, contact with herding cultures, history of rebellions, latitude and longitude.

Q5. The transferability of PRSs developed for Europeans to other populations is controversial. The authors should assess the prediction performance of the used PRSs in the investigated study population, individual information on height, BMI, education attainment and, particularly important due to the results, age at first birth should be available. The general paragraph "recent studies suggest that the results can well apply to East Asian (e.g., Chinese) populations. For example, Duncan and colleagues (34) demonstrated that the polygenic score performance is reasonably reliable in East Asian samples (95%) relative to European samples (100%)." is not enough.

Response: Thanks for pointing this out. We agree with the reviewer that ideally, the Chinese-specific polygenic scores should be created and used (e.g., based on GWAS data from people of Chinese ancestry), or the exact prediction accuracy of each polygenic trait in the Chinese population should be tested. Unfortunately, due to budget/resource constraints and lack of existing literature, we are unable to perform the Chinese-specific GWAS or acquire the exact prediction performance of each polygenic score among the Chinese population in the current study. In the revised draft, we add this major research limit in the discussion section on page 19.

Q6. Please replace Fig. 2 by scatter plots with the same axes and (non)-linear regression lines if needed.

Response: Thanks for the helpful suggestion. As the reviewer suggested, we plot the average polygenic scores against the % of paddy rice levels as in Figure R3, and replace Fig. 2 in the revised manuscript.

Figure R3. Average Polygenic Scores by County Rice Paddy Percentage

Q7. For testing reverse causality, I would suggest simply to interchange the response and explanatory variables in the OLS and 2SLS analyses.

Response: Thanks for pointing this out. We agree with the reviewer that it is possible that the start of rice farming about 9,400 years ago may be explained by different genetic endowments of ancient people, i.e., the reverse causality. Although 2SLS estimation results presented in the original manuscript can partially alleviate the concern of reverse causality, there is currently no paleoanthropological evidence or data (e.g., ancient DNA) that can be used to directly test or completely rule out such concern. We also consider the reviewer's suggestion to interchange the response and explanatory variables in the 2SLS analysis, but in that case, the current instrumental variable used (i.e., the environmental suitability for wetland rice) would not be a suitable IV of polygenic scores or genotypes (e.g., fail to meet with the "relevance" criteria of a valid instrumental variable). In the revised draft, we add this major research limit in the discussion section on page 20.

References:

- Chen, J., Zheng, H., Bei, J. X., Sun, L., Jia, W. H., Li, T., ... & Liu, J. (2009). Genetic structure of the Han Chinese population revealed by genome-wide SNP variation. *The American Journal of Human Genetics*, 85(6), 775-785.
- Gomez, J. C., & Moens, M. F. (2012). PCA document reconstruction for email classification. *Computational Statistics & Data Analysis*, 56(3), 741-751.
- Li, B., Cai, X., Wang, L., Li, J., Zou, Y., Chen, G., & Wang, S. (2021). A Genome-Wide Association Study Finds Variants at 2p21 Associated with Self-Reported Sensitive Skin in the Han Chinese population. *The Journal of investigative dermatology*, S0022-202X.
- Peterson, R. E., Edwards, A. C., Bacanu, S. A., Dick, D. M., Kendler, K. S., & Webb, B. T. (2017). The utility of empirically assigning ancestry groups in cross-population genetic studies of addiction. *The American journal on addictions*, 26(5), 494-501.
- Perneger, T.V., 1998. What's wrong with Bonferroni adjustments. *BMJ*, 316(7139), pp.1236-1238.
- Romano, J.P. and Wolf, M., 2005. Stepwise multiple testing as formalized data snooping. *Econometrica*, 73(4), pp.1237-1282.
- Sankoh, A.J., Huque, M.F. and Dubey, S.D., 1997. Some comments on frequently used multiple endpoint adjustment methods in clinical trials. *Statistics In Medicine*, 16(22), pp.2529-2542.